# The antibacterial effect of human adipose-derived stem cells on LL-37-resistant bacteria

**Parisa Afzal Haghjoo[1], Ali Mojtahedi[2], Malek Moien Ansar** [ORCID]**[3,4]\*, Malek Masoud Ansar[5,6], Safieh Danesh Mobarhan[5]**

**1** Department of Microbiology, School of Medicine, Guilan Univercity of Medical Sciences, Rasht, Guilan, Iran, **2** Department of Microbiology, School of Medicine, Iran University of Medical Sciences, Tehran, Tehran, Iran, **3** Neuroscience Research Center, Trauma Institute, Guilan University of Medical Sciences, Rasht, Guilan, Iran, **4** Department of Biochemistry, School of Medicine, Guilan University of Medical Sciences, Rasht, Guilan, Iran, **5** Department of Anatomy, School of Medicine, Guilan University of Medical Sciences, Rasht, Guilan, Iran **6** Cellular and Molecular Research Center, Guilan University of Medical Sciences, Rasht, Guilan, Iran

\* ansar_moien@yahoo.com

## Abstract

Human adipose-derived stem cells (hADSCs) exhibit antibacterial properties, but their effectiveness against bacteria resistant to LL-37- a natural human antimicrobial peptide important in the immune defense- is not fully understood. Some bacteria have evolved mechanisms to evade the antimicrobial effects of LL-37. We aimed to investigate the antibacterial efficacy of hADSCs against *Pseudomonas aeruginosa, Proteus mirabilis, and methicillin-resistant Staphylococcus aureus* (MRSA), focusing on the antimicrobial peptide LL-37. hADSCs were isolated from human adipose tissue, identified by flow cytometry and differentiation assays, and divided into three groups: unstimulated, stimulated with interferon-gamma (IFN-γ; 100 ng/mL) or *Escherichia coli* (300 CFU). LL-37 gene expression was measured by qPCR after 6 hours in the *E. coli* stimulated group. LL-37 peptide levels were quantified by ELISA in conditioned media from unstimulated, IFN-γ stimulated cells, both before and after incubation with pathogens (300 CFU). Antibacterial activity was assessed by colony counting incubation following incubation of conditioned media with bacteria. Conditioned media from both unstimulated and stimulated hADSCs significantly inhibited growth of all three pathogens ($P < 0.05$), with highest efficacy against *P. aeruginosa* (86.4% inhibition), followed by MRSA (74%) and *P. mirabilis* (63%). LL-37 gene expression increased after bacterial stimulation, and also LL-37 concentrations increased in conditioned media but significantly decreased after bacterial exposure ($P < 0.05$). Despite this reduction, antibacterial activity persisted. hADSC-conditioned media exert potent antibacterial effects against LL-37-resistant pathogens, even when LL-37 levels are reduced after bacterial exposure. These findings support the therapeutic potential of hADSC secretomes, particularly for infections caused by bacteria capable of reducing LL-37 levels.

**Data availability statement:** All relevant data are within the manuscript and its Supporting Information files.

**Funding:** This research was supported by Guilan University of Medical Sciences. The sponsors had no role in study design, data collection and analysis, decision to publish, or preparation of the manuscript. No funding was received to cover publication costs.

**Competing interests:** The authors have declared that no competing interests exist.

**Abbreviations: AMPs**, antimicrobial peptides; **CFU**, Colony-Forming Unit; **DMEM**, Dulbecco's Modified Eagle Medium; **ECM**, hADSCs stimulated by E. coli; **FBS**, fetal bovine serum; **hADSCs**, human adipose-derived stem cells; **ICM**, hADSCs stimulated by interferon gamma; **IDT**, Integrated DNA Technologies; **IF**N-γ, interferon-gamma; **MRSA**, methicillin-resistant Staphylococcus aureus; **MSCs**, mesenchymal stromal/stem cells; **PBS**, phosphate-buffered saline; **qPCR**, Quantitative Real-Time PCR

## Introduction

Infectious diseases caused by bacteria, viruses, or parasites are responsible for many deaths worldwide. Due to the global rise in drug-resistant infections, there has been increasing interest in developing new treatments and approaches [1,2]. Recently, the potential of stem cells to treat infections has gained attention [3]. One probable mechanism underlying the antimicrobial effects of mesenchymal stromal/ stem cells (MSCs) is the secretion of antimicrobial peptides (AMPs) [4,5]. AMPs exhibit selective activity against a wide range of organisms, including bacteria, yeasts, fungi, viruses, and even cancer cells [4]. The best-known human antimicrobial peptide is LL-37- the only member of human Cathelicidin family- which is found in various tissues and body fluids and can be secreted by different cell types [6] including MSCs [7–9]. The antimicrobial effects of LL-37 are mediated by direct interactions with microbial cell membrane, leading to disruption of membrane integrity and the immunomodulatory effects on inflammatory cells [10]. Despite recent research, studies on the antibacterial effects of human adipose-derived stem cells (hADSCs) remain limited [11]. It has been shown that different stem cell types may have distinct antimicrobial effects depending on both the cell type and the pathogen [7,9]. Furthermore, many studies have reported that LL-37 is the main mediator of antibacterial action of MSCs [7] but some bacteria secrete proteases that may degrade LL-37.

The present study investigated the antibacterial effect of hADSCs against three clinically relevant strains: Methicillin-resistant *Staphylococcus aureus* (MRSA), *Proteus mirabilis* and *Pseudomonas aeruginosa.* MRSA continues to be a major multidrug-resistant pathogen causing serious infections worldwide, with rising concerns over antibiotic resistance [12]. *P. mirabilis* resists the antimicrobial peptide LL-37 primarily through secretion of proteases such as ZapA that degrade this peptide [13]. *P. aeruginosa* is a notorious opportunistic pathogen, largely due to its biofilm formation and intrinsic resistance mechanisms that complicate treatment [14]. Recent studies have shown that hADSCs secrete LL-37 and other antimicrobial factors, contributing to their broad antibacterial potential [15].

This study focused on assessing the antibacterial effects of hADSCs on LL-37-resistant bacteria, including *P. mirabilis* and MRSA [13,16], as well as *P. aeruginosa* under *in vitro* conditions.

## Materials and methods

### Ethics and sample collection

This study was approved by the Guilan University Ethics Committee (Ethics code: IR.GUMS.REC.1396.62) and conducted in accordance with the Declaration of Helsinki and institutional ethical guidelines. Participants were recruited between 30 August 2017 and 10 January 2018. Written informed consent was obtained and all procedures adhered to relevant guidelines and regulations.

Adipose tissue samples were collected from four unrelated healthy subjects undergoing liposuction or adipose tissue dissection for weight loss. Active standard bacterial strains used for antibacterial assays were procured from the Pasteur Institute and the National Center for Genetic and Biological Resources, Iran.

## Isolation and culture of hADSCs

Mesenchymal stem cells were isolated from adipose tissue samples using the enzymatic digestion method described by Bunnell et al. [17]. Briefly, adipose tissue was washed with phosphate-buffered saline (PBS), mechanically minced, and incubated in 0.25% collagenase type I solution for 40 minutes at 37°C. The cell suspension was centrifuged at 1200 rpm for 10 minutes to collect the cell pellet. Collagenase activity was neutralized by adding Dulbecco's Modified Eagle Medium (DMEM) supplemented with 10% fetal bovine serum (FBS) and penicillin-streptomycin. The pellet was resuspended in the same medium and seeded into tissue culture flasks, designated as passage 0, and incubated at 37°C in a humidified atmosphere containing 5% $CO_2$. The culture medium was refreshed every three days.

When cells reached approximately 70–80% confluency, they were rinsed twice with PBS, detached using trypsin-EDTA, and incubated at 37°C for 2 minutes. Trypsin activity was neutralized by adding culture medium. After centrifugation at 1200 rpm for 5 minutes, the cell pellet was resuspended and split into three flasks for further expansion.

## Cryopreservation and experimental design

Isolated stem cells were cryopreserved following established protocols to maintain viability and functionality [18]. The main experiments were conducted after cryopreservation to ensure consistent cell viability and functionality, allowing for standardized timing and reproducibility of assays.

## Flow cytometry analysis

Cells at passage 2 were used for flow cytometry analysis [8]. Approximately 30,000 cells in 50 µL PBS were incubated with 2 µL of phycoerythrin (PE)- or fluorescein isothiocyanate (FITC)-conjugated antibodies against CD90, CD105, CD73, CD34 and CD45. After washing with PBS, the pellet was resuspended in 500µl PBS and transferred to flow cytometry tubes. Data were analyzed using FloMax software.

## Adipogenic and osteogenic differentiation assay

For differentiation assays, 10,000 cells per well (adipogenic) and 6000 cells per well (osteogenic) were seeded in a 24-well plates. After 24 hours, the culture medium was replaced with 1 ml of differentiation medium. The adipogenic medium consisted of DMEM high glucose supplemented with 10% FBS, $10^{-7}$ M dexamethasone, 0.5 mM IBMX, 66 nM insulin, and 0.2 mM indomethacin, The osteogenic medium consisted DMEM high glucose supplemented with 10% FBS, $10^{-7}$ M dexamethasone, and 50 µg/ml Ascorbic acid bi- phosphate. The differentiation medium was changed every three days. After 21 days, cells were fixed with 10% formalin for 1 hour at room temperature and stained with Oil Red O to detect lipid droplets. On day 15, cells were fixed and stained with Alizarin Red to detect calcium deposits. Cells from passage 3 onwards were used for these assays.

## Preparation of supernatant and conditioned medium from hADSCs

Cells from passages 4–9 were used for experiments [8]. Medium from non-stimulated adipose-derived stem cells was prepared by seeding 200,000 cells per well in 24-well plate and culturing for 24 hours. The medium was then replaced with antibiotic-free medium. After an additional 24 hours, the medium was collected, centrifuged, and the clear supernatant stored in a −20° C [19, 20]. For conditioned medium from stem cells stimulated by *Escherichia coli*: bacteria were cultured overnight in LB broth at 37° C with gentle agitation and then plated on LB agar. After 24 hours, fresh bacterial colonies were dissolved in 2 ml of PBS and incubated for 3 hours. A bacterial suspension at 0.5 McFarland standard was prepared, diluted to 300 CFU/30 µL with PBS, and added to 470 µl of DMEM containing 10% FBS in wells containing 200,000 stem cells at 37° C. After 6 hours, the medium was collected, centrifuged, filtered, and stored at −20° C. For medium from stem cells stimulated with interferon-gamma (IFN-γ), cells were incubated with 500 µL DMEM containing 10% FBS and IFN-γ at concentration of 100ng/µL (or 2000 U/µL) [6].

## Antimicrobial activity

Four bacterial standard strains were tested: *E. coli* ATCC 25922, *P. aeruginosa* ATCC 27853, *P. mirabilis* ATCC 7002 and MRSA ATCC 33591. Bacteria were cultured overnight in BHI broth at 37 °C with gentle agitation and then plated on blood agar. After 24 hours, fresh colonies were suspended in 2 ml of LB broth and adjusted to 0.5 McFarland standard. The suspension was diluted to 300 CFU/30 µL with LB broth. In a 24-well plate, 570 µL of each collected medium were incubated with 30µl of bacterial suspension for 6 hours at 37° C. Bacterial counts were determined by serial dilution, and inhibition rates were compared to control. Gentamicin was used as the positive control, with serial two-fold dilutions (final concentration 100 to 0.39 µg/ml). Each well was inoculated with 300 CFU of bacteria and incubated for 6 hours. Gentamicin-treated samples were plated on appropriate media (LB agar for *E. coli and P. aeruginosa*, EMB agar for *P. mirabilis*, and blood agar for MRSA). Colony counts were performed after 24 hours. The effective gentamicin concentrations for *P. aeruginosa*, *P. mirabilis,* and MRSA were 6.25, 3.12, and 12.5 µg/mL, respectively.

## Quantitative real-time PCR (qPCR)

ADSCs (passages 4–9) were seeded at $2 \times 10^5$ cells per well in 24-well plates with complete DMEM containing 10% FBS. After 24 hours, the medium was replaced with fresh DMEM supplemented with 300 CFU *E. coli*, with or without 1.25 µg/mL heparin; untreated cells served as controls. Following 6 hours of incubation, total RNA was extracted using RNX-Plus, quantified by NanoDrop, and treated with DNase I to eliminate genomic DNA contamination. cDNA synthesis was performed using the 2X RT PRE-MIX Kit. Specific primers for LL-37 — forward 5′-TAACCTCTACCGCCTCCTGGACCT GGACC-3′ and reverse 5′-GGACTCTGTCCTGGGTACAAGATTCCGC-3′ — and GAPDH — forward 5′-CAAGATCAT CACCAATGCCT-3′ and reverse 5′-CCCATCACGCCACAGTTTCC-3′ — were designed via Integrated DNA Technologies (IDT) and validated by agarose gel electrophoresis. Quantitative PCR was carried out on a Roche LightCycler using BioFACT™ 2X SYBR Green Master Mix under cycling conditions of 94°C for 10 minutes, followed by 40 cycles of 94°C for 20 seconds and 72°C for 20 seconds; annealing temperatures were set at 60°C for LL-37 and GAPDH. All reactions were performed in duplicate, and relative gene expression levels were determined using the 2^(-ΔΔCt) method normalized to GAPDH and control samples.

## LL-37 measurements

Levels of human LL-37 in collected media were measured using an ELISA kit (Zellbio, Germany, Cat No: ZB-12197S-H9648), following the manufacturer's instructions.

## Statistical analysis

Data were analyzed using SPSS 22 software (IBM, USA). Experiments were performed in duplicate using ADSCs from four unrelated healthy donors. Results are presented as mean± SEM. The Mann-Whitney U test was used to compare two groups. For comparisons among multiple groups, one-way ANOVA with Tukey's post hoc test was performed. $P < 0.05$ was considered statistically significant.

## Results

### Morphological characteristics of hADSCs

After initial seeding, stem cells adhered to the flask surface within 2–3 days and displayed a fibroblast-like morphology Figure a in S1 Fig. Approximately ten days later, these fibroblast-like cells persisted and reached a confluency of approximately 70–80% Figure b in S1 Fig.

## Flow cytometry analysis of different CD markers

hADSCs expressed high levels of MSC markers. The expression levels of CD73, CD90, and CD105 were 97.96±1.33%, 98.27±1.18%, and 97.75±0.89%, respectively. The expression of CD34 (0.92±0.46%) and CD45 (0.41±0.22%) was low, indicating the absence of endothelial and hematopoietic markers S2 Fig.

## Adipogenic and osteogenic differentiation

The cells differentiated into adipocytes and osteocytes under specific differentiation conditions. Adipogenic differentiation was confirmed by intracellular lipid droplets stained with Oil Red O Figure a and b in S3 Fig, and osteogenic differentiation was indicated by mineralization, visualized by Alizarin Red staining Figure c and d in S3 Fig.

## LL-37 expression and secretion

LL-37 expression in hADSCs increased by 2.54±0.759fold following 6 hours of *E. coli* stimulation compared to control, as determined by qPCR Fig 1, indicating rapid transcriptional activation. Correspondingly, LL-37 peptide concentrations in conditioned media increased after 24 hours of culture with or without IFN-γ stimulation, as determined by ELISA. However, LL-37 protein levels decreased following incubation with bacteria, likely due to bacterial protease degradation (F (7, 47) = 8.5, P=0.000). Post-hoc analysis showed significant reductions with *P. aeruginosa* and *P. mirabilis* (P<0.05 and P<0.01 for unstimulated and IFN-γ-stimulated cells, respectively), and with MRSA at P<0.01 for both conditions Fig 2. These results demonstrate that LL-37 gene induction precedes peptide secretion, and secreted LL-37 is subject to proteolytic reduction in the presence of pathogens.

## Antimicrobial activity of hADSCs

Growth of *P. aeruginosa* (Fig 3a), *P. Mirabilis* (Fig 3b) and MRSA (Fig 3c) in hADSCs-conditioned medium from *E. coli*-stimulated cells was reduced compared to the controls (P<0.05). The percentage of growth inhibition was: *P. aeruginosa* (86.4%), *P. Mirabilis* (63%), and MRSA (74%). Additionally, both unstimulated and IFN-γstimulated hADSCs showed significant inhibition of all three bacteria compared to the controls (F (3, 22) =9.88, P=0.000, F (3, 20) = 11.31, P=0.000, F (3, 19) = 13.29, P=0.000, respectively) Fig 4a–4c.

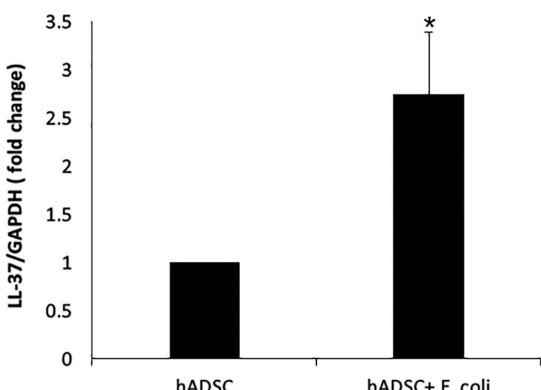

**Fig 1. qPCR LL-37 gene expression.** Relative LL-37 gene expression normalized to GAPDH in ADSCs and *E. coli*-stimulated. All reactions were performed in duplicate. Data are mean± SEM. ADSC: adipose-derived stem cells; *E. coli*: 300 CFU; • P<0.05 compared to ADSC group.

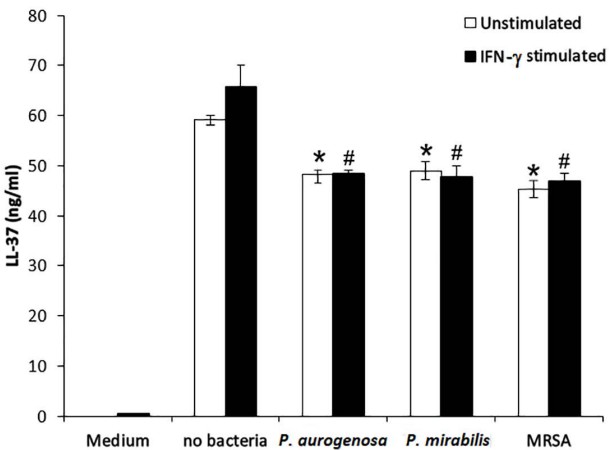

**Fig 2. LL-37 concentration.** LL-37 values were shown in medium obtained from non-stimulated hADSCs and stimulated by interferon gamma. Both mediums incubated without bacteria, *P. aeruginosa*, *P. mirabilis* and MRSA separately. 570 µl of each collected medium was incubated with 300 CFU of three bacteria (in 30 µl of PBS) and incubated for 6 h. All media were collected and the levels of LL-37 were measured. The data represent measurements using hADSCs from four healthy unrelated donors and are presented as mean± SEM. (n: 7 to 11) * P<0.05 versus no bacteria (unstimulated) # P<0.01 versus no bacteria (IFN-γ stimulated). * P<0.01, CFU: Colony-Forming Unit.

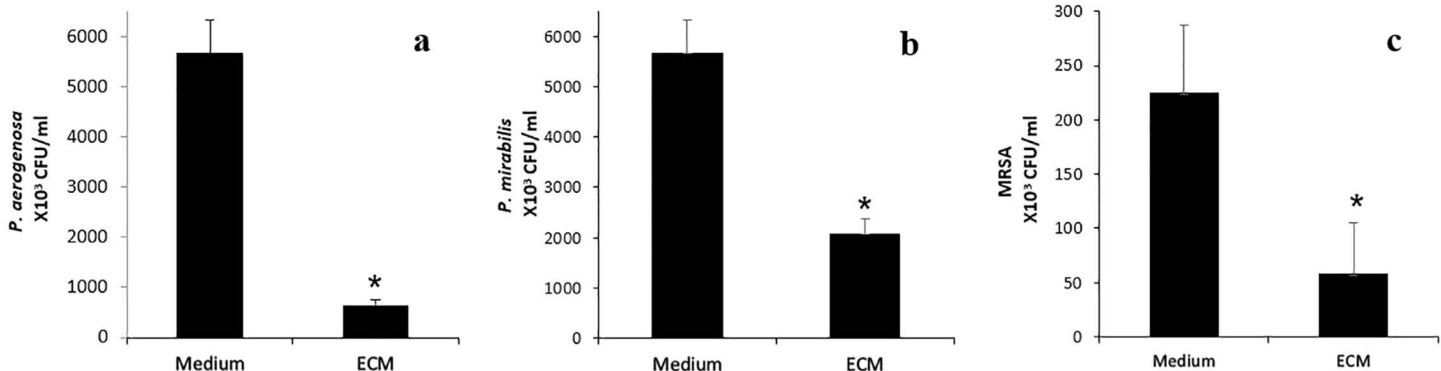

**Fig 3. Antibacterial activity of medium derived from hADSCs stimulated by *E. coli* (ECM) on the growth of *P. aeruginosa*, *P. mirabilis* and MRSA.** (a) 570µl of each collected medium was incubated with 300 CFU of *P. aeruginosa* (in 30µl of PBS). (b) 570 µl of each collected medium was incubated with 300 CFU of *P. mirabilis* (in 30 µl of PBS) (c) 570 µl of each collected medium was incubated with 300 CFU of MRSA (in 30 µl of PBS). All samples were incubated for 6 h, and bacterial counts in each well determined by serial dilution. The data represent measurements using hADSCs from four healthy unrelated donors and are presented as mean± SEM. (n: 7 to 11). * P<0.05, CFU: Colony-Forming Unit;.

## Discussion

The results demonstrated that the conditioned media from hADSCs significantly inhibited the growth of MRSA, *P. mirabilis, and P. aeruginosa* with the highest inhibition observed against *P. aeruginosa* (86.4%), followed by MRSA (74%) and *P. mirabilis* (63%). Despite a significant reduction in LL-37 peptide levels in conditioned media after bacterial exposure [12,13]—likely due to bacterial protease activity—antibacterial activity persisted, indicating that LL-37 is not the sole mediator of this effect. Quantitative PCR analysis revealed that LL-37 gene expression in hADSCs increased approximately 2.54-fold after 6 hours of *E. coli* stimulation, confirming that LL-37 production is inducible in response to bacterial challenge.

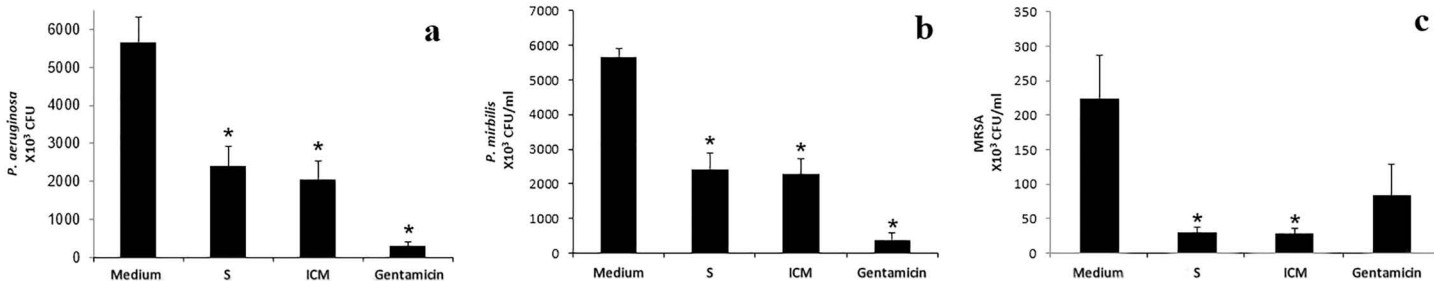

**Fig 4. Antibacterial activity of medium derived from non-stimulated hADSCs (S) and stimulated by interferon gamma (ICM) on the growth of *P. aeruginosa*, *P. mirabilis* and *MRSA*.** (a) 570 μl of each collected medium was incubated with 300 CFU of *P. aeruginosa* (in 30 μl of PBS), b) 570μl of each collected medium was incubated with 300 CFU of *P. mirabilis* (in 30μl of PBS), c) 570 μl of each collected medium was incubated with 300 CFU of MRSA (in 30 μl of PBS) incubated for 6 h. Bacterial counts in each well determined by serial dilutions. The data represent measurements using hADSCs from four healthy unrelated donors and are presented as mean± SEM. (n: 7 to 11). Gentamicin was used as positive control at a concentration of 6.25 μg/ml. * P < 0.05, CFU: Colony-Forming Unit.

The measurement of LL-37 gene expression by qPCR after 6 hours of *E. coli* exposure captures early transcriptional responses of hADSCs to bacterial challenge. This rapid induction precedes detectable increases in secreted peptide levels measured by ELISA after 24 hours of culture with or without IFN-γ stimulation. The observed decrease in LL-37 protein following bacterial incubation reflects proteolytic degradation by bacterial enzymes known to degrade antimicrobial peptides [12–14]. Together, these complementary assays elucidate the kinetics of LL-37 production and degradation, highlighting the dynamic interplay between stem cell antimicrobial responses and bacterial evasion strategies.

This dual measurement approach strengthens the mechanistic understanding of hADSC antimicrobial activity by linking gene induction with peptide secretion and degradation dynamics. It also supports the conclusion that, despite protease-mediated LL-37 reduction, additional secreted factors contribute to the sustained antibacterial effects of hADSC-conditioned media.

Previous studies have demonstrated that the antibacterial activity of both stimulated and non-stimulated BMSCs is primarily mediated by the secretion of LL-37 [8,9], with one study reporting increased LL-37 gene expression exclusively in stimulated cells [8]. Among eight studies reporting LL-37 secretion by MSCs [11], only one focused on hADSCs and examined LL-37 expression following vitamin D exposure [21]. Conversely, Miranda et al. showed that LL-37 gene expression was not induced in stimulated BMSCs or menstrual stem cells by lipopolysaccharide or bacterial exposure, identifying hepcidin as the major antibacterial agent instead [20].

LL-37 secretion significantly increased in both unstimulated and IFN-γ-stimulated hADSCs. Previous studies on bone marrow-derived MSCs have often emphasized LL-37 as the primary antibacterial effector; however, our findings and limited prior research on hADSCs suggest tissue-specific differences in antimicrobial profiles. For example, while synthetic LL-37 is ineffective against *P. mirabilis* due to protease-mediated degradation [12], hADSC-conditioned media demonstrated stronger antibacterial activity, implying synergistic effects of multiple secreted factors.

Conditioned media from unstimulated and IFN-γ-stimulated hADSCs were used for our experiments, while media from *E. coli*-stimulated cells were excluded due to bacterial protease-mediated degradation of LL-37, which interferes with accurate quantification [22]. This approach provided clearer insights into the dynamics of LL-37 expression and degradation, reinforcing the potential of hADSC secretomes as cell-free antimicrobial agents.

The current results demonstrated that, despite a significant reduction in LL-37 levels following exposure to *P. aeruginosa*, *P. mirabilis* and MRSA exposure, conditioned media from both stimulated and unstimulated hADSCs maintained strong antibacterial activity, highlighting the contribution of multiple antimicrobial peptides and factors beyond LL-37 [4,8]. Given the similar effects observed across these bacterial strains, we proceeded to analyze and interpret the responses of each strain separately.

*P. aeruginosa* is a common nosocomial pathogen and the second most frequent cause of ventilator-associated pneumonia [23]. Its ability to form biofilms is critical for survival. LL-37 is the only cathelicidin known to possess anti-biofilm activity [24]. In the present study, the antibacterial effects against *P. aeruginosa* persisted despite reduced LL-37 levels, suggesting that residual LL-37 or other antimicrobial factors secreted by hADSCs contribute to bacterial inhibition. Notably, P. *aeruginosa* showed the greatest growth inhibition among tested bacteria in hADSCs-conditioned media. This supports previous finding that *P. aeruginosa* is sensitive to LL-37 despite production of proteases like elastase that degrade LL-37 [25]. Our data reveal that hADSC-conditioned media retain potent activity against *P. aeruginosa* despite reduced LL-37 levels, supporting the hypothesis that residual LL-37 and other secreted antimicrobial components act in concert to inhibit bacterial growth. Therefore, hADSC-conditioned media, even without stimulation, may be a potential therapeutic agent against *P. aeruginosa* infections.

To our knowledge, this is the first report demonstrating the antimicrobial activity of hADSCs-conditioned media against *P. mirabilis* and implicating LL-37 in this effect. *P. mirabilis* is resistant to synthetic LL-37 [12,26], likely due to ZapA, an extracellular matrix metalloprotease that degrades antimicrobial peptides such as hBD1 and LL-37 [12]. A decrease was observed in LL-37 levels in conditioned media exposed to *P. mirabilis*, yet the antibacterial activity of conditioned media remained stronger than that reported for synthetic LL-37 alone [12]. This suggests that (a) LL-37 is partially degraded by ZapA; (b) residual LL-37 is still effective, and (c) other antibacterial factors secreted by hADSCs contribute to the activity. The enhanced potency of ADSCs-derived LL-37 likely reflects its natural context and synergy with additional antimicrobial agents in the conditioned media.

*P. mirabilis* causes symptomatic urinary tract infections including cystitis and pyelonephritis, which may progress to complications such as bacteremia and potentially life-threatening urosepsis [27]. Given the antibacterial potency of hADSC-conditioned media and LL-37 secretion, these media, even without additional stimulation, may represent a promising therapeutic approach for such infections.

The antimicrobial effects of hCAP18/LL-37, LL-37 and recombinant Cathelin on MRSA ATCC 33591 have shown that this strain is resistant to the peptides hCAP18/LL-37 and LL-37 peptides, but remains sensitive to Cathelin [16]. The present findings demonstrated that hADSCs effectively reduced MRSA (ATCC 33591), despite a decrease in LL-37 concentration. This reduction may be attributed to the ability of MRSA to secrete proteases. Previous studies have identified several proteases secreted by MRSA including aureolysin, V8 protease, staphopain A (ScpA) and ScpB. Notably, ScpB has been reported to degrade LL-37 at concentrations higher than those observed in our study [28,29]. To date, only two studies have investigated the interaction between MRSA and stem cells: one utilized BMSCs and focused on LL-37 expression without quantifying its concentration [30], while the other examined the effect of hADSC secretome on MRSA without characterizing the specific secreted factors involved [15].

Given the high incidence of MRSA, accounting for up to 74% of S. aureus infections worldwide [31], the emergence of vancomycin-resistant MRSA strains, and adverse antibiotic effects, further research is warranted to hADSC antimicrobial activity against MRSA and to elucidate underlying mechanisms.

While this study focused on *in vitro* experiments to delineate fundamental antibacterial mechanisms, these findings provide a strong rationale for future *in vivo* studies to evaluate the therapeutic potential, safety, and pharmacokinetics of hADSC secretomes in relevant infection models. Additionally, further characterization of the non-LL-37 antimicrobial components in hADSC secretions is warranted to fully understand and optimize their clinical application.

## Conclusion

This study demonstrates that hADSC significantly inhibit the growth of multidrug-resistant MRSA, LL-37-resistant *P. mirabilis*, and clinically relevant *P. aeruginosa*. Upon bacterial stimulation, LL-37 gene expression in hADSCs is inducible, and conditioned media from hADSC - stimulated with or without IFN-γ- show enhanced LL-37 secretion and strong antibacterial activity. Notably, despite a reduction in LL-37 concentration, the conditioned media maintain robust antibacterial effect, indicating that may residual LL-37 or possibly other secreted factors secreted by hADSCs contribute to this activity. These

finding underscore the antimicrobial potential of hADSC secretomes and support their development as promising alternative or adjunctive therapies against resistant and difficult-to-treat bacterial infections, especially those capable of reducing LL-37 levels. Future *in vivo* studies are essential to validate these results, further elucidate the underlying mechanisms, and assess the therapeutic efficacy and safety of hADSC- derived treatments.

## Supporting information

**S1 Fig. Morphological impacts of human stem cells derived from adipose tissue.** a) Proliferation on days 2–3 χ 20, b) Cells with fibroblast-like shape and 70–80% confluency χ10.
(TIF)

**S2 Fig. Flow cytometry analysis of ADSCs markers expression.** Scatter plots showed the ADSCs were positive for CD34-CD45 (a), CD73 (b) and were negative for CD90 (c) and CD105 (d).
(TIF)

**S3 Fig. Adipocyte Stem cell differentiation capacity in adipogenic and osteogenic differentiation medium.** (a) Differentiated cells without staining. (b) Differentiated cells stained by Oil Red O. (c) Differentiated cells without staining. (d) Differentiated cells stained by Alizarin Red.
(TIF)

**S1 Appendix. Raw FCS data for FC_cd45_cd34_adsc1.** Raw flow cytometry standard file for CD 45 and CD 34 markers analyzed in ADSCs from donor 1.
(FCS)

**S1 Text. Metadata for FC_cd45_cd34_adsc1.** Experimental details and acquisition parameters.
(DOCX)

**S2 Appendix. Raw FCS data for FC_cd73_adsc1.** Raw flow cytometry standard file for CD 73 marker analyzed in ADSCs from donor 1.
(FCS)

**S2 Text. Metadata for FC_cd73_adsc1.** Experimental details and acquisition parameters.
(DOCX)

**S3 Appendix. Raw FCS data for FC_cd105_cd90_adsc1.** Raw flow cytometry standard file for CD 105 and CD 90 markers analyzed in ADSCs from donor 1.
(FCS)

**S3 Text. Metadata for FC_cd105_cd90_adsc1.** Experimental details and acquisition parameters.
(DOCX)

**S4 Appendix. Raw FCS data for FC_cd45_cd34_adsc2.** Raw flow cytometry standard file for CD 45 and CD 34 markers analyzed in ADSCs from donor 2.
(FCS)

**S4 Text. Metadata for FC_cd45_cd34_adsc2.** Experimental details and acquisition parameters.
(DOCX)

**S5 Appendix. Raw FCS data for FC_cd73_adsc2.** Raw flow cytometry standard file for CD 73 marker analyzed in ADSCs from donor 2.
(FCS)

**S5 Text.  Metadata for FC_cd73_adsc2.** Experimental details and acquisition parameters.
(DOCX)

**S6 Appendix.  Raw FCS data for FC_cd90_adsc2.** Raw flow cytometry standard file for CD 90 marker analyzed in ADSCs from donor 2.
(FCS)

**S6 Text.  Metadata for FC_cd90_adsc2.** Experimental details and acquisition parameters.
(DOCX)

**S7 Appendix.  Raw FCS data for FC_cd105_adsc2.** Raw flow cytometry standard file for CD 105 marker analyzed in ADSCs from donor 2.
(FCS)

**S7 Text.  Metadata for FC_cd105_adsc2.** Experimental details and acquisition parameters.
(DOCX)

**S8 Appendix.  Raw FCS data for FC_cd45_cd34_adsc3.** Raw flow cytometry standard file for CD 45 and CD 34 markers analyzed in ADSCs from donor 3.
(FCS)

**S8 Text.  Metadata for FC_cd45_cd34_adsc3.** Experimental details and acquisition parameters.
(DOCX)

**S9 Appendix.  Raw FCS data for FC_cd73_adsc3.** Raw flow cytometry standard file for CD 73 marker analyzed in ADSCs from donor 3.
(FCS)

**S9 Text.  Metadata for FC_cd73_adsc3.** Experimental details and acquisition parameters.
(DOCX)

**S10 Appendix.  Raw FCS data for FC_cd90_adsc3.** Raw flow cytometry standard file for CD 90 marker analyzed in ADSCs from donor 3.
(FCS)

**S10 Text.  Metadata for FC_cd90_adsc3.** Experimental details and acquisition parameters.
(DOCX)

**S11 Appendix.  Raw FCS data for FC_cd105_adsc3.** Raw flow cytometry standard file for CD 105 marker analyzed in ADSCs from donor 3.
(FCS)

**S11 Text.  Metadata for FC_cd105_adsc3.** Experimental details and acquisition parameters.
(DOCX)

**S12 Appendix.  Raw FCS data for FC_cd45_cd34_adsc4.** Raw flow cytometry standard file for CD 45 and CD 34 markers analyzed in ADSCs from donor 4.
(FCS)

**S12 Text.  Metadata for FC_cd45_cd34_adsc4.** Experimental details and acquisition parameters.
(DOCX)

**S13 Appendix. Raw FCS data for FC_cd73_adsc4.** Raw flow cytometry standard file for CD 73 marker analyzed in ADSCs from donor 4.
(FCS)

**S13 Text. Metadata for FC_cd73_adsc4.** Experimental details and acquisition parameters.
(DOCX)

**S14 Appendix. Raw FCS data for FC_cd90_adsc4.** Raw flow cytometry standard file for CD 90 marker analyzed in ADSCs from donor 4.
(FCS)

**S14 Text. Metadata for FC_cd90_adsc4.** Experimental details and acquisition parameters.
(DOCX)

**S15 Appendix. Raw FCS data for FC_cd105_adsc4.** Raw flow cytometry standard file for CD 105 marker analyzed in ADSCs from donor 4.
(FCS)

**S15 Text. Metadata for FC_cd105_adsc4.** Experimental details and acquisition parameters.
(DOCX)

**S1 Dataset. Raw qPCR data for Fig. 1.** Raw cycle threshold (Ct) values for LL-37 and GAPDH gene expression measured in ADSCs and *E. coli*-stimulated ADSCs. Data were used to calculate relative LL-37 gene expression shown in Fig 1.
(XLSX)

**S1 File. Metadata for qPCR assay.** Experimental metadata including details on sample preparation, qPCR reagents and kits, primer sequences for LL-37 and GAPDH, reaction conditions, and technical duplicate measurements.
(DOCX)

**S2 Dataset. Raw LL-37 ELISA data for Fig 2.** Concentrations of LL-37 measured by ELISA in culture medium obtained from unstimulated and IFN-g stimulated hADSCs. Medium samples were incubated without bacteria or with *P. aeruginosa*, *P. mirabilis*, or MRSA separately. Data represent measurements using hADSCs from four healthy unrelated donors.
(XLSX)

**S2 File. Metadata for LL-37 ELISA assay.** Experimental metadata including bacterial strains used for incubation and replicates.
(DOCX)

**S3 Dataset. Raw data for Fig 3a.** Individual CFU counts of *P. aeruginosa* after incubation with hADSC medium stimulated by *E. coli*, Data represent measurements using hADSCs from four healthy unrelated donors and were used to generate Fig 3a.
(XLSX)

**S3 File. Metadata for antibacterial assay.** Experimental metadata including bacterial strain and CFU counts and replicates.
(DOCX)

**S4 Dataset. Raw data for Fig 3b.** Individual CFU counts of *P. mirabilis* after incubation with hADSC medium stimulated by *E. coli*, Data represent measurements using hADSCs from four healthy unrelated donors and were used to generate Fig 3b.
(XLSX)

**S4 File. Metadata for antibacterial assay.** Experimental metadata including bacterial strain and CFU counts and replicates.
(DOCX)

**S5 Dataset. Raw data for Fig 3c.** Individual CFU counts of MRSA after incubation with hADSC medium stimulated by *E. coli*, Data represent measurements using hADSCs from four healthy unrelated donors and were used to generate Fig 3c.
(XLSX)

**S5 File. Metadata for antibacterial assay.** Experimental metadata including bacterial strain and CFU counts and replicates.
(DOCX)

**S6 Dataset. Raw data for Fig 4a.** Individual CFU counts of *P. aeruginosa* after incubation with medium derived from unstimulated and IFN-g stimulated hADSCs. Data represent measurements using hADSCs from four healthy unrelated donors and support Fig 4a.
(XLSX)

**S6 File. Metadata for antibacterial assay.** Experimental metadata including bacterial strain and CFU counts and replicates.
(DOCX)

**S7 Dataset. Raw data for Fig 4b.** Individual CFU counts of *P. mirabilis* after incubation with medium derived from unstimulated and IFN-g stimulated hADSCs. Data represent measurements using hADSCs from four healthy unrelated donors and support Fig 4a.
(XLSX)

**S7 File. Metadata for antibacterial assay.** Experimental metadata including bacterial strain and CFU counts and replicates.
(DOCX)

**S8 Dataset. Raw data for Fig 4c.** Individual CFU counts of MRSA after incubation with medium derived from unstimulated and IFN-g stimulated hADSCs. Data represent measurements using hADSCs from four healthy unrelated donors and support Fig 4a.
(XLSX)

**S8 File. Metadata for antibacterial assay.** Experimental metadata including bacterial strain and CFU counts and replicates.
(DOCX)

## Acknowledgments

Special thanks are dedicated to Mr. Mostafa Haghiri (MA in English linguistics) for editing the manuscript

## Author contributions

**Conceptualization:** Ali Mojtahedi, Malek Moien Ansar, Malek Masoud Ansar.

**Data curation:** Ali Mojtahedi, Malek Moien Ansar.

**Formal analysis:** Malek Moien Ansar.

**Funding acquisition:** Malek Moien Ansar.

**Investigation:** Parisa Afzal Haghjoo, Ali Mojtahedi, Malek Masoud Ansar, Safieh Danesh Mobarhan.

**Methodology:** Parisa Afzal Haghjoo, Ali Mojtahedi, Malek Moien Ansar.

**Project administration:** Malek Moien Ansar.

**Supervision:** Malek Moien Ansar, Malek Masoud Ansar, Ali Mojtahedi.

**Validation:** Malek Moien Ansar.

**Visualization:** Malek Moien Ansar.

**Writing – original draft:** Parisa Afzal Haghjoo, Safieh Danesh Mobarhan.

**Writing – review & editing:** Ali Mojtahedi, Malek Moien Ansar, Malek Masoud Ansar.

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
