## [Decision Letter · Decision Letter 0]

7 Aug 2025

PONE-D-25-35482The antibacterial effect of human adipose-derived stem cells on LL-37-resistant bacteriaPLOS ONE

Dear Dr. Ansar,

Thank you for submitting your manuscript to PLOS ONE. After careful consideration, we feel that it has merit but does not fully meet PLOS ONE’s publication criteria as it currently stands. Therefore, we invite you to submit a revised version of the manuscript that addresses the points raised during the review process.

We look forward to receiving your revised manuscript.

Kind regards,

Christian Agyare, PhD

Academic Editor

PLOS ONE

Additional Editor Comments:

Please act and respond to the reviewers comments and submit the revised version for our consideration

Reviewers' comments:

Reviewer's Responses to Questions

**Comments to the Author**

1. Is the manuscript technically sound, and do the data support the conclusions?

Reviewer #1: Yes

Reviewer #2: Yes

2. Has the statistical analysis been performed appropriately and rigorously? 

Reviewer #1: Yes

Reviewer #2: Yes

3. Have the authors made all data underlying the findings in their manuscript fully available?

Reviewer #1: Yes

Reviewer #2: Yes

4. Is the manuscript presented in an intelligible fashion and written in standard English?

Reviewer #1: Yes

Reviewer #2: Yes

5. Review Comments to the Author

Reviewer #1: Greetings

Good work

Minor points:

1- In this study: You wrote (We 7 times and Our 9 times!). The rule of scientific

manuscripts’ writing is: it is preferred to avoid using the pronouns (We and

Our). So you should delete (We and Our), kindly use the formal scientific

words (This study or The current study or The present study).

2- kindly you should write all the scientific names of bacteria by italic style line.

3- The abbreviations in this manuscript are too much, so I suggest to add table of abbreviations at the end of this manuscript to be easy for understanding by the students or the readers.

Kind regards

Reviewer #2: Overall english language and the scientific approach used in this study are good.

However, there are some points needs to be ammended or answered. especially, The abstract needs to be rephrased to increase clarity for readers and some statements needs more supportive referencing in discussion session. I've attched my notes on the original pdf.

Regards

6. PLOS authors have the option to publish the peer review history of their article (what does this mean? ). If published, this will include your full peer review and any attached files.

**Do you want your identity to be public for this peer review?** For information about this choice, including consent withdrawal, please see our Privacy Policy .

Reviewer #1: No

Reviewer #2: No

---

## [Author Response · Author response to Decision Letter 1]

13 Sep 2025

Response to Editor and Reviewers’ Comments

Dear Editor and Reviewers,

We sincerely appreciate the time and effort you have dedicated to reviewing our manuscript. We have thoroughly addressed all comments and suggestions to improve the clarity, rigor, and compliance of our work with PLOS ONE guidelines. Below is our point-by-point response to each comment, along with a summary of specific changes made in the revised manuscript.

Response to Editor’s Comments

Manuscript Formatting and Style

We carefully reviewed and reformatted the manuscript in accordance with the official PLOS ONE style requirements for the main text, title page, authors, and affiliations. This included file naming, section headings, consistent font sizes, and reference style to fully comply with journal requirements.

Citation of Suggested Publications

We thank the reviewers for suggesting that additional references be added in specific section to support certain statements. Although no particular publications were recommended, we carefully reviewed the manuscript and inserted relevant citations where appropriate to strengthen the discussion.

Reference List Accuracy and Retraction Status

We have conducted a thorough review of our reference list and, to the best of our knowledge, none of the cited articles have been retracted. DOIs for two references were verified, corrected, and formatted according to PLOS ONE style.

Figures and PACE Tool Usage

All figures were prepared according to PLOS ONE guidelines using the PACE (PLOS Article Content Editing) tool to ensure correct resolution, dimensions, and font embedding. Figures were uploaded separately as required, improving quality and compliance.

Response to Reviewer 1 Comments

Use of Pronouns

We revised the manuscript to minimize use of personal pronouns such as “We” and “Our.” Formal alternatives such as “This study,” “The present study,” or “The current study” is now used consistently.

Italicization of Scientific Names

All scientific names of bacterial species have been updated and italicized throughout the manuscript, consistent with taxonomic convention (e.g., P. aeruginosa, P. mirabilis).

Abbreviation Table

The abbreviation table was added based on Reviewer 1’s suggestion to aid reader comprehension. While PLOS ONE guidelines do not explicitly require such a table, we included it to address the reviewer’s helpful recommendation.

Response to Reviewer 2 Comments

Rephrasing:

The indicated sentence was rephrased to improve clarity and flow.

Clarification of the compound:

We added the phrase “a natural human antimicrobial peptide important in the immune defense” and the sentence “Some bacteria have evolved mechanisms to evade the antimicrobial effects of LL-37” to clearly identify and explain the compound.

“Identified by” or “Differentiated by”:

We selected the term “identified by” to convey a precise and unambiguous description of the methods used.

Clarification of groups and stimulation:

We have added the sentence: “Divided into three groups: unstimulated, stimulated with interferon-gamma (IFN-γ; 100 ng/mL), or Escherichia coli (E. coli; 300 CFU),” and inserted “in the E. coli-stimulated group” in the appropriate section to clarify group distinctions.

Clarification of “IDT”:

“Integrated DNA Technologies (IDT)” has been introduced upon first mention.

Font size issue:

The font size in the Statistical analysis subsection of the Methods was increased to 12 pt for clarity and consistency with journal standards.

References missing or insufficient:

References 12, 13, and 14 were added where needed to provide appropriate support.

Italicization of species names and formatting adjustments:

Species names were corrected to italic font (e.g., P. aeruginosa [5 cases], P. mirabilis [4 cases]). The “Discussion” heading was reformatted in bold with an18-pt font size to match PLOS One style.

If any additional modifications or clarifications are needed, we will be pleased to respond promptly. We thank the Editor and Reviewers once again for their valuable feedback and hope that the revised manuscript meets the standards for publication.

Respectfully,

Dr Malek Moien Ansar

---

## [Editor Report · Decision Letter 1]

17 Sep 2025

The antibacterial effect of human adipose-derived stem cells on LL-37-resistant bacteria

PONE-D-25-35482R1

Dear Dr. Ansar,

We’re pleased to inform you that your manuscript has been judged scientifically suitable for publication and will be formally accepted for publication once it meets all outstanding technical requirements.

Kind regards,

Christian Agyare, PhD

Academic Editor

PLOS ONE
---

## [Editor Report · Acceptance letter]

PONE-D-25-35482R1

PLOS ONE

Dear Dr. Ansar,

I'm pleased to inform you that your manuscript has been deemed suitable for publication in PLOS ONE. Congratulations! Your manuscript is now being handed over to our production team.

Kind regards,

on behalf of

Dr. Christian Agyare

Academic Editor

PLOS ONE